# DEEP AUDIO PRIOR

## ABSTRACT

Deep convolutional neural networks are known to specialize in distilling compact and robust prior from a large amount of data. We are interested in applying deep networks in the absence of training dataset. In this paper, we introduce deep audio prior (DAP) which leverages the structure of a network and the temporal information in a single audio file. Specifically, we demonstrate that a randomly-initialized neural network can be used with carefully designed audio prior to tackle challenging audio problems such as universal blind source separation, interactive audio editing, audio texture synthesis, and audio co-separation.

To understand the robustness of the deep audio prior, we construct a benchmark dataset *Universal-150* for universal sound source separation with a diverse set of sources. We show superior audio results than previous work on both qualitative and quantitative evaluations. We also perform thorough ablation study to validate our design choices. `https://iclr-dap.github.io/Deep-Audio-Prior/`

## 1 INTRODUCTION

Typically, a deep neural network distills robust priors from a large amount of labeled or unlabeled data (Deng et al., 2009; Jansen et al., 2018). In audio research, neural networks such as VGGish (Hershey et al., 2017) and Wave-U-Net (Stoller et al., 2018) have shown great success in audio classification, audio source separation, and many other challenging tasks. While large audio datasets have greatly improved supervised training, the collection and especially the cleaning of a large amount of audio data still remain an open challenge (Fonseca et al., 2017). For example, in AudioSet (Gemmeke et al., 2017), one of the popular audio datasets, we often find audios/videos that contain content beyond what the label specifies[1].

In this paper, we combine the power of deep neural networks and temporal prior in audio without any external training data. Similar ideas have been explored in deep image prior (DIP) (Ulyanov et al., 2018). Double-DIP from Gandelsman et al. (2019) further showed that it is possible to achieve robust unsupervised image decomposition from a single-image input, without pre-training on any data. Deep Audio Prior (DAP)'s capability to train on a single audio file has several advantages. First, with proper selection of the audio priors, we show that DAP generalizes well to a wide variety of unseen types of data. Second, our training process is fully unsupervised and therefore make it possible to pre-process large volumes of data in the wild. Last but not least, we show several novel applications that are only possible because of the unique features of DAP, including universal source separation, interactive editing, audio texture synthesis, and audio co-separation.

Domain gap between *audio* and *visual images* precludes direct adoption of the image priors. Many assumptions or priors that are true for images no longer hold for audio. By nature, audio signals exhibit strong temporal coherence, e.g. one's voice changes smoothly (See $A \to B$ in inset). Since images tend to have more spatial patterns, most existing deep image priors have focused on how to encapsulate the spatial redundancy. Another challenge specific to audio is the activation discontinuity. Unlike in videos where an object moves continuously in the scene, a sound source can make sound or turn complete silence at any given time (see $A \to C$ in inset).

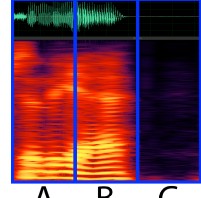

Coherence and discontinuity in audio.

Our proposed deep audio prior framework has the following main contributions:

- A temporally coherent source generator that can reproduce a wide spectrum of natural sound sources including human speech, musical instruments, animal sounds, etc.

---

[1]As an example, under "bark" from AudioSet, we find `https://youtu.be/2mJbGx5D-zA?t=150` containing human speech, wind noises, and various audio effects other than bark.

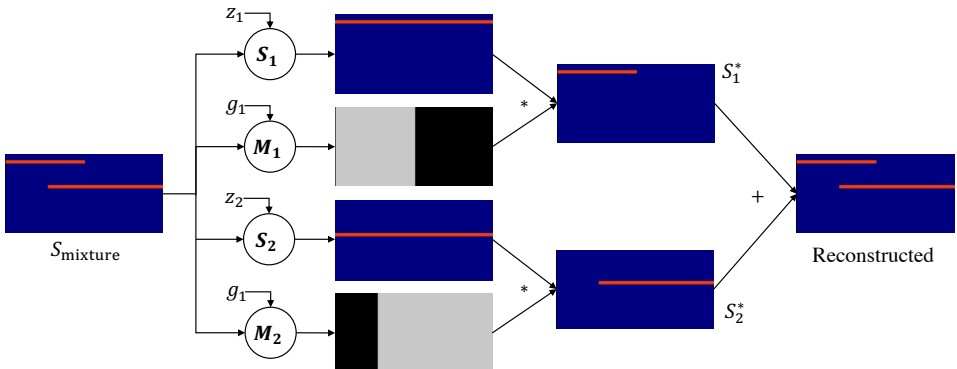

Figure 1: DAP framework illustrated with a synthetic sound mixture spectrogram. With random noises as inputs, we use two sound prediction networks: $S_1$ and $S_2$ and two mask modulation networks: $M_1$ and $M_2$ to perform source separation. The four networks share a same U-Net structure (Ulyanov et al., 2018).

- A novel mask activation scheme to enable and disable sources *with* frequency binning and *without* temporal dependence.

- We demonstrated the effectiveness of DAP in several challenging applications, including universal audio source separation, interactive mask-based editing, audio texture synthesis, and audio co-separation.

- We also introduce Universal-150, a benchmark dataset on blind source separation for universal sound sources. We showed that DAP outperforms other blind source separation (BSS) methods in multiple numerical metrics.

## 2 DEEP AUDIO PRIOR FRAMEWORK

Deep Audio Prior is an unsupervised blind source separation framework. More specifically, DAP does not train on any extra data other than the input audio mixture. Similar to image foreground and background segmentation, audio blind two-source separation can be expressed as:

$$S_{\text{mixture}} = S_1(z_1) * M_1(g_1) + S_2(z_2) * M_2(g_1) \tag{1}$$

where $S_1$ and $S_2$ are two audio generator networks, $M_1$ and $M_2$ are two mask networks, $z_1, z_2$ and $g_1, g_2$ are sampled from random distributions. Our method works in Time-Frequency (T-F) spectrogram space (Cohen, 1995). All the input and output variables are of dimension $F \times T$. Waveforms are transformed into spectrograms by Short-Time Fourier Transform (STFT) with a frame size of 1022 and hop size of 172, and audio sampling rate is 11000 Hz. Figure 1 shows an overview of our framework.

The intuition for our single-audio source separation is inspired by single-image decomposition: it is much easier for two generator networks ($S_1$ and $S_2$) to learn two distinct sound sources respectively, rather than forcing one of the generators to learn the mixture. Gandelsman et al. (2019) analyzed this in terms of complex mixture versus simple individual components. In audio, as shown in §3.1, we show that networks tend to quickly learn patterns from two distinct sources.

**Reconstruction Loss**  Let $S_1^* = S_1(z_1) * M_1(g_1)$ and $S_2^* = S_2(z_2) * M_2(g_2)$, then we have $S_{\text{mixture}}^* = S_1^* + S_2^*$. Clearly, when combining the separated sounds $S_1^*$ and $S_2^*$, we should obtain the original sound mixture $S_{\text{mixture}}$. The data fidelity term is expressed as a reconstruction loss, $l_{\text{rec}}$, which will push the combination of the two separated sounds to be close to the original sound mixture.

$$l_{\text{rec}} = ||S_{\text{mixture}} - S_1^* - S_2^*||_2. \tag{2}$$

## 2.1 Temporally Coherent and Dynamic Source Generator

**Temporally Coherent Source Generation** Sound signals from the same source across time would be similar (Rosen, 1992). To explicitly model the temporal property, we use multiple audio frames with temporal consistent noise as inputs for each sound source. We split the $S_{\text{mixture}}$ into $N$ frames along the time axis and obtain $\{S_{\text{mixture}}^i\}_{i=1}^N$. Accordingly, we use $n$ noise input pairs $\{z_1^i, z_2^i\}_{i=1}^N$ to predict the corresponding sounds $\{S_1(z_1^i), S_2(z_2^i)\}_{i=1}^N$. For the rest of this paper, we will omit the underscript when there is no confusion, i.e. $z^i$ instead of $z_1^i$.

To explicitly enforce the temporal coherence, we impose strong correlations on input noises:

$$z^1 = n^* \quad \text{and} \quad z^i = z^{i-1} + \Delta u^i \tag{3}$$

where $\Delta u^i$ is a random noise sampled from an uniform distribution with a variance significantly lower than that of Gaussian noise $n^*$ initialization of $z^1$. Since we use shared networks $S_1$ and $S_2$ across different frames for predicting individual sounds and the noise inputs are temporally consistent, which will enforce the network to predict temporal consistent sounds. A similar idea was also adopted in (Gandelsman et al., 2019) to preserve video coherence but they use the noise to predict masks. We also employ a temporal continuity loss function to further enforce it by pushing the absolute of gradients along time to be small (Chambolle, 2004):

$$l_{\text{tc}} = \sum_i \sum_p \sum_q |S_i(p, q) - S_i(p, q - 1)|. \tag{4}$$

**Dynamic Source Generation** Audio signals can also have dynamic patterns with large variations. Since values in $z^i - z^{i-1} = \Delta u^i$ are very small by construction, the temporally consistent noise can only handle small variations and is not capable of capturing temporally dynamic sound patterns. To preserve the temporally consistent patterns in predictions and also hallucinate dynamic patterns, in spirit of curriculum learning (Bengio et al., 2009), we gradually add dynamic noise into inputs as we progress more training iterations:

$$z^i = \alpha(t)z^{i-1} + \Delta u^i + (1 - \alpha(t))n^i \tag{5}$$

where the $t$ denotes optimization iteration, and $n^i$ refers to a random Gaussian noise. Unlike the temporal consistent noise in (3) that have a constant $n^*$ as initialization, $n^i$ is independently sampled for each frame $i$. To balance the $n^*$ and $n^i$ throughout our training iterations, we introduce a coefficient $\alpha(t)$:

$$\alpha(t) = \begin{cases} 1 & \text{if } t < T_1 \quad \cdots \quad \text{initially } z^i \text{ are very smooth,} \\ \frac{T_2 - t}{T_2} & \text{if } T_1 \leq t \leq T_2 \quad \cdots \quad \text{gradually transit from smooth to dynamic inputs,} \\ 0 & \text{if } t > T_2 \quad \cdots \quad \text{at the end } z^i \text{ are sufficiently dynamic,} \end{cases} \tag{6}$$

where $T_1$ and $T_2$ are two thresholds and $T_2 > T_1$. If $t < T_1$, only a temporally shared constant noise $n^*$ is used; if $T_1 \leq t < T_2$, the dynamic sampled noise $n^i$ will be gradually added; if $t > T_2$, only a dynamic noise is used. The noise input design will first let the model predict stable temporally consistent sound patterns and then push the model to capture large variations in spectrograms to reduce the reconstruction loss. Note that $\alpha(t)$ is not continuous with respect to $t$ (see red curve in inset). In practice, we found this abrupt change in $\alpha(t)$ improves the ability to capture dynamic audio change. See more discussion in §4.

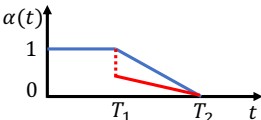

Sharp change in red better captures dynamic variation.

**Frequency Domain Exclusion** We assume that two different sounds $S_1^*$ and $S_2^*$ are dissimilar. To enforce the constraint, we utilize an exclusion loss function from Zhang et al. (2018) in which it is formulated as the product of normalized gradient fields of the two sound predictions at $L$ spatial resolutions:

$$l_{\text{ex}} = \sum_{n=1}^L ||\Psi(S_1^{*(n)}, S_2^{*(n)})||_F \quad \text{and} \quad \Psi(x, y) = \tanh(\lambda_1|\nabla x|) \odot \tanh(\lambda_2|\nabla y|), \tag{7}$$

where $\lambda_1$ and $\lambda_2$ are regularizers, $n$ the spatial patch index, $|| \cdot ||_F$ the Frobenius norm, $\odot$ denotes element-wise multiplication. We set $L = 3$, $\lambda_1 = \sqrt{\frac{||\nabla y||_F}{||\nabla x||_F}}$, and $\lambda_2 = \sqrt{\frac{||\nabla x||_F}{||\nabla y||_F}}$.

## 2.2 1D MASK CONSTRAINTS FOR AUDIO

Sound sources will not always make sounds all the time, which would break the temporal consistency in the spectrogram domain. To address this issue, we introduce audio masks to activate sounding spectrum regions and deactivate silent regions.

If a source is sounding at a time, the spectrum bin at the time should be activated. Based on the observation, mask values within the same temporal bin should be consistent and binary. Namely, if a dog barking sound is present, it should appear across all frequency range under the same timestamp. Therefore, we force the mask within the same temporal bin to be consistent: $M_i(0, q) = M_i(1, q) = ... = M_i(F - 1, q) = m_i^q$, where $i$ refers to $i$-th sound source, $q \in \{0, 1..., T\}$, $F$ and $T$ are height (frequency axis) and width (time axis) of the spectrogram $M$. In our implementation, we use a max pooling operator to aggregate the mask content along frequency-coordinate and generate mask value $m_i^q$ using a Sigmoid function. $m_i^q$ represents the mask for $i$-th source at time $q$. We define $m_i^q$ to be a continuous variable for ease of optimization and will enforce an extra binary loss term.

**Non-Zero Masks**    Furthermore, at any time, if there is a sound in $S_{\text{mixture}}$, at least one of the masks should be activated. To this end, we introduce a nonzero mask loss term:

$$l_{\text{nonzero}} = \sum_q w_q \cdot (\epsilon + \min(\sigma, \sum_i m_i^q))^{-1} \quad \text{and} \quad w_q = \sum_p \log(1 + S_{\text{mixture}}(p, q)), \quad (8)$$

where $\epsilon = 10^{-6}$ for numerical stability and $\sigma = 1$ is a margin value. The margin value is to ensure when the sum of mask activation is already larger than the $\sigma$, the loss will not continue to push the mask values. $w^q$ is to suppress this loss term if there is no sound in $S_{\text{mixture}}$ at time $q$. Moreover, in our observations, we found that masks are either fully activated or fully deactivated. Very rarely would a sound source be at $50\%$ activation level. Therefore, we introduce a differential loss term to encourage the mask networks to generate binary masks: $l_{\text{binary}} = \sum_i(\epsilon + \sum_{p,q} |M_i(p, q) - 0.5|)^{-1}$, where again $\epsilon = 10^{-6}$ is used to avoid numerical issues. The $l_{\text{binary}}$ will force the mask values to be as far away as possible to $0.5$, which means they are close to either $0$ or $1$.

## 2.3 WALKTHROUGH ON SYNTHETIC SEPARATION EXAMPLES

With all the pieces together, our total loss is defined as:

$$l_{\text{total}} = l_{\text{rec}} + l_{\text{tc}} + l_{\text{ex}} + l_{\text{nonzero}} + l_{\text{binary}}. \quad (9)$$

We optimize the networks in an end-to-end manner with all the loss functions. We empirically keep the weights for all loss terms the same, with the only exception being $l_{\text{binary}}$ has a $0.01$ factor. To validate the implementation of our algorithm, we generate two types of synthetic spectrograms.

**(i) Single-frequency band fake sounds**    We generate a synthetic mixture where each of the two audio sources is producing a flat tone at a single frequency. For this test, since we know the spectrogram at different segment will always produce the same output (each sound source is just a flat bar), we set the input noise $z_1^1 = z_1^2 = \cdots = z_1^n$ for sound source 1 and the same for source 2. The top two rows from Figure 2 show that our DAP achieves perfect prediction on both the source generators and masks.

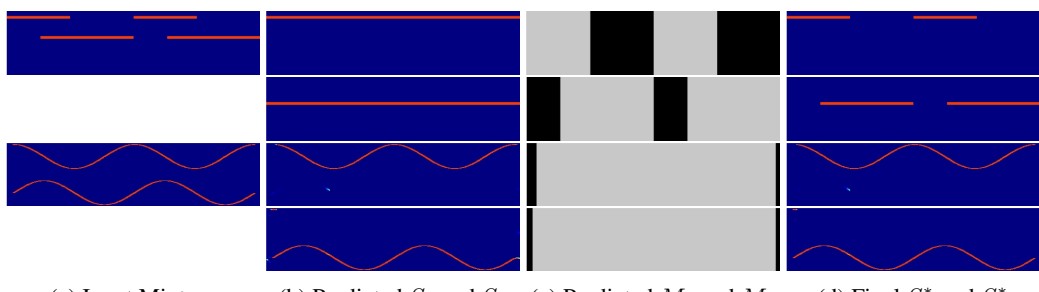

(a) Input Mixture    (b) Predicted $S_1$ and $S_2$   (c) Predicted $M_1$ and $M_2$    (d) Final $S_1^*$ and $S_2^*$

Figure 2: Synthetic tests: two single-tone sources (top two rows) and two cosine-tone sources (bottom two rows). Given single input mixture, DAP achieves perfect separation on both inputs.

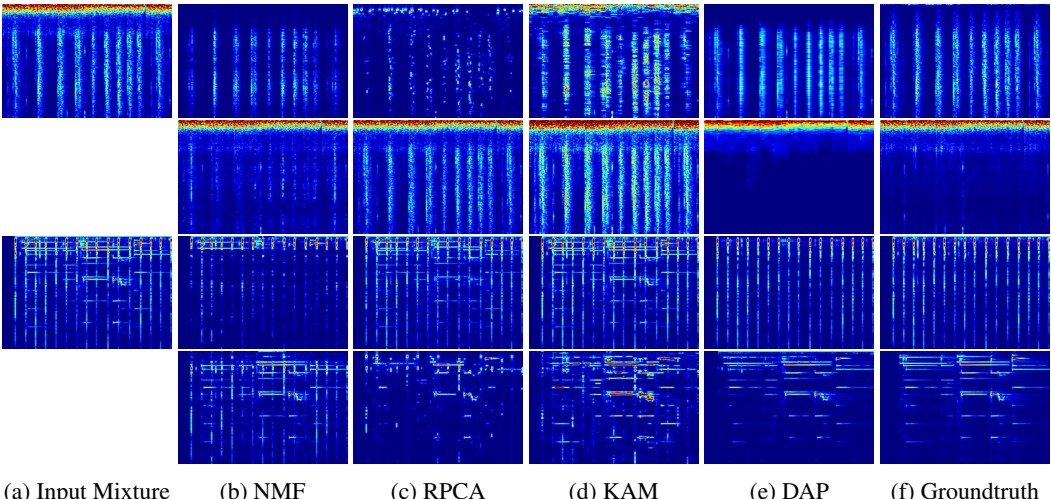

| (a) Input Mixture | (b) NMF | (c) RPCA | (d) KAM | (e) DAP | (f) Groundtruth |

Figure 3: Audio comparison on Universal-150 benchmark. Qualitatively, DAP significantly outperforms all the other blind separation methods. For full results, please check out our project page: `https://iclr-dap.github.io/Deep-Audio-Prior/`

**(ii) Curved input sounds**  Our next test is to validate if our DAP framework can handle temporally coherent spectrogram changes. Two shifted cosine curves are combined together as input to our separation pipeline (see bottom two rows in Figure 2). We set the input noise according to equation (3). Again, our DAP framework achieves the desired separation of sources and masks.

## 3   APPLICATIONS

Next, we present several challenging applications with DAP. For the best experience, please use Adobe PDF Reader[2] or visit our anonymous webpage[3] to listen to our audio results. Unless specified otherwise, all the experiments for the same application are run with the same set of parameters.

### 3.1   UNIVERSAL BLIND SOURCE SEPARATION

Given a sound mixture $S_{\text{mixture}}$, universal blind source separation aims to separate individual sounds from the sound mixture without using any external data. Universal blind separation is challenging because the input audio can be in arbitrary domain, not just commonly studied speech or music domains.

**Universal-150 Audio Benchmark**  To the best of our knowledge, there is no publicly available *universe audio source separation dataset*. Therefore, we built such a dataset that contains 150 audio mixtures and each sample is a two-sound mixture. These mixture samples come from pairs of 30 unique sounds from YouTube and ESC50 sound classification dataset (Piczak, 2015) covering a large range of sound categories appeared in our daily life, such as animal (*e.g.,* dog, cat, and rooster), human (*e.g.,* human speech, baby crying, and baby laughing), music (*e.g.,* violin and guitar), natural sounds (*e.g.,* rain, sea wave, and crackling fire), domestic and urban sounds (*e.g.,* clock, keyboard typing, and siren).

**Comparison with Blind Separation Methods**  We compare our method with several blind source separation (BSS) methods: non-negative matrix factorization (NMF) from Spiertz & Gnann (2009), robust principal component analysis (RPCA) from Huang et al. (2012), and kernel additive modelling (KAM) from Yela et al. (2018). Figure 3 shows that our method outperforms the compared methods qualitatively. Please either use Adobe PDF Reader and click on each spectrogram or visit our webpage to listen to the audios. For implementations, we use (Manilow et al., 2018) for NMF and RPCA and (Yela et al., 2018) for KAM.

---

[2]In Adobe PDF Reader, click on each spectrogram image and enable the audio plugin when prompted.

[3]Our anonymous submission page: `https://iclr-dap.github.io/Deep-Audio-Prior/`

Table 1: Comparison between DAP/NMF/RPCA/KAM on numerical metrics: SDR/SIR/LSD. For SDR/SIR, higher is better and for LSD, smaller is better. The best results on the three metrics are from our method, DAP.

|  | NMF (Spiertz & Gnann, 2009) | RPCA (Huang et al., 2012) | KAM Yela et al. (2018) | Proposed DAP |
|---|---|---|---|---|
| SDR | -6.369 | -5.395 | -3.375 | **-1.581** |
| SIR | -1.934 | -0.227 | -0.753 | **3.859** |
| LSD | 2.301 | 2.011 | 2.321 | **1.959** |

Moreover, to quantitatively evaluate sound separation performance of different BSS methods, we run separation on all 150 sounds and compare these methods in three metrics: Signal-to-Distortion Ratio (SDR), Signal-to-Interference Ratio (SIR), and Audio Spectrum Distance (LSD) (Vincent et al., 2006; Morgado et al., 2018). The SDR and SIR measure the distortion and interferences in the separated audios. LSD measures the Euclidean distance between a predicted audio spectrum magnitude and the corresponding ground truth audio spectrogram magnitude.

DAP outperforms these three BSS methods in all three numerical metrics, as shown in Table 1. Note that all these four methods, including DAP, do not require any training data. The only input to the algorithm is the single mixture audio file.

**Comparison with Methods using Deep Networks**    Since existing deep audio separation networks were usually trained on music or speech data, they are supervised and can not handle unseen sounds in our universe sound separation dataset. So we compare our method with two state-of-the-art deep audio networks: Deep Network Prior (DNP) from Michelashvili & Wolf (2019) and Speech Enhancement Generative Adversarial Network (SEGAN) from Pascual et al. (2017) on noisy speech data. Figure 4 illustrates speech denoising results. For supervised model, in our tests, SEGAN works well on noises that are similar to the training set. However, for unseen novel noises like the keyboard typing noise in Figure 4 (a), SEGAN did not remove the noise. The proposed DAP can remove background noise. A side effect of DAP aggressively removing noises is the excessive removal of speech signals, which might lead to lower quality in some cases. Note that we do not impose any speech-related denoising prior in the universal DAP model.

### 3.2    Interactive Mask-based Editing

Since the output of our method contains both a mask and the sound generator, we can add additional constraints on either the mask or the sound spectrogram. Traditionally this type of interaction is only performed on spectrograms since they don't have the predicted masks available (Bryan et al., 2014). There are several drawbacks with using spectrogram strokes: spectrograms are non-intuitive for non-audio professionals, and even then, a lot of strokes are needed for real-world audios.

In comparison, constraints on 1D masks are a lot easier for users to specify. Figure 5 shows the simple 1D box given by the user can quickly improve the results. These 1D box constraints can be easily drawn by users that are not familiar with frequency spectrograms. Basically they select

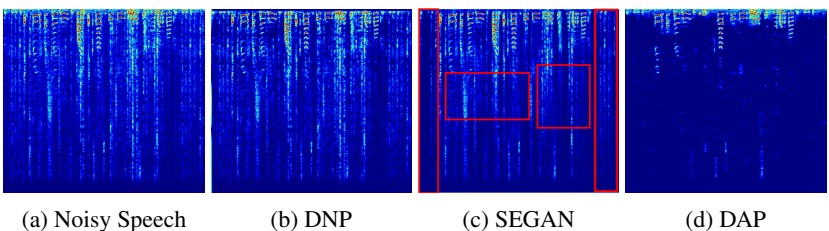

| (a) Noisy Speech | (b) DNP | (c) SEGAN | (d) DAP |
|---|---|---|---|

Figure 4: Comparison with other deep network models. DNP does not require training dataset and DAP achieve higher denoising quality. As for supervised models, for example SEGAN, that are trained on large dataset, although they usually work well on seen noises, it is hard to generalize to unseen noise. Here the speech is mixed with a noise that was not in the training set. While SEGAN did not remove the novel noise, DAP can better remove noises at the price of lower speech quality in some segments. Please listen to the sounds to appreciate the difference.

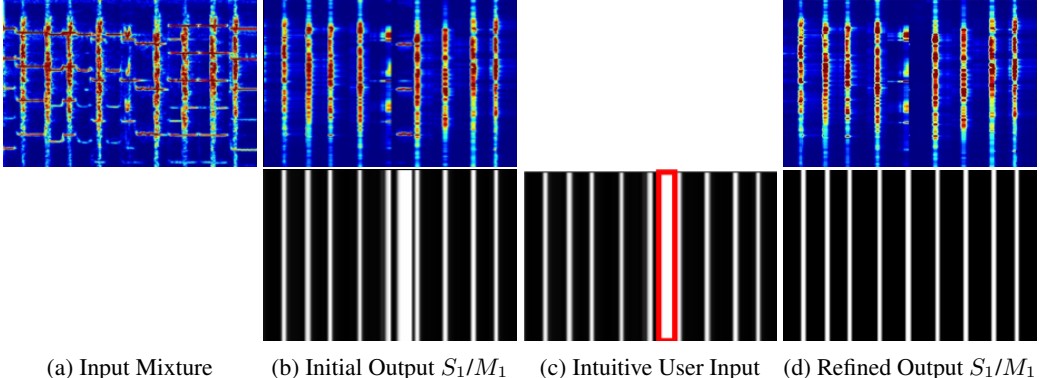

| (a) Input Mixture | (b) Initial Output $S_1/M_1$ | (c) Intuitive User Input | (d) Refined Output $S_1/M_1$ |

Figure 5: Example of interactive DAP. Given a sound mixture, our DAP model can predict separated sounds and the corresponding mask activation maps. Users can simply draw boxes to interact with our DAP model to tell where to deactivate or activate the predicted masks for refining predicted sounds. As shown, a user deactivate a region in the predicted mask for a separated sound, and then we obtain better results with the refined mask. All spectrograms are zoomed in for visualization.

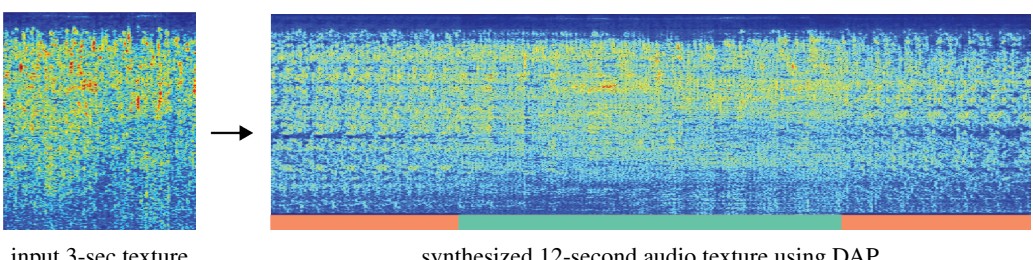

input 3-sec texture    synthesized 12-second audio texture using DAP

Figure 6: Audio texture synthesis via latent space augmentation. From a 3-second input audio, we can interpolate (in green) and extrapolate (in orange) the latent input noise to synthesize a seamless 12-second audio.

the regions they think should have sound or should be silence for source $k$. These selected regions would become input activation masks: $M_k^{\text{act}}$ or deactivate mask: $M_k^{\text{deact}}$, respectively. The values in the annotated regions are one. To encode these annotations, we introduce mask activation and deactivation losses to refine results from our separation networks:

$$l_{\text{act}} = \sum_k M_k^{\text{act}} \|M_k - M_k^{\text{act}}\|_1 \quad \text{and} \quad l_{\text{deact}} = \sum_k M_k^{\text{deact}} \|M_k - (1 - M_k^{\text{deact}})\|_1. \quad (10)$$

With the activation and deactivation losses, our DAP framework refines the source generator *and* masks. This refinement process typically takes tens of seconds. Figure 5 shows an interactive mask editing result. We can see that with adding a mask deactivation loss for the "dog" sound into optimization, our network will remove the "violin" patterns from it.

### 3.3 AUDIO TEXTURE SYNTHESIS

Another interesting application is audio synthesis or audio interpolation/extrapolation. The goal of audio synthesis is to lengthen a given audio. This has wide applications in audio texture generation for arbitrary length to match visual content. Here we show an example on how we can leverage the input noise latent space to achieve audio texture synthesis (Figure 6).

Given an audio texture, we apply DAP framework and obtain the temporally coherent input noises $[z^1, z^2, ..., z^n]$ for the whole sequence and a generator $S$. To lengthen the input texture, we can first use straightforward *interpolation*: we can insert more noise frame between every two consecutive noise frames: $\boldsymbol{z}_{int} = [z^1, \boxed{z^{1.5}}, z^2, \boxed{z^{2.5}}, ..., \boxed{z^{n-0.5}}, z^n]$. Those symbols in boxes are new interpolated ones – this way we can easily double (or more) the length of the input audio. If we want to explore more diversity in our learned generator, we can also extrapolate in the latent space.

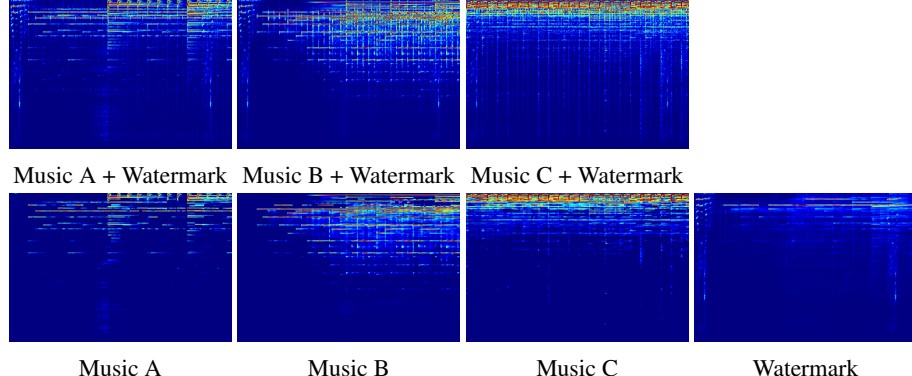

| Music A + Watermark | Music B + Watermark | Music C + Watermark | |
| Music A | Music B | Music C | Watermark |

Figure 7: DAP can also be applied to automatically remove audio watermark. Top row shows three input mixture audios A, B, and C, all mixed with watermark. Given these 3 mixtures, DAP can seperate each individual sound source out. Note that DAP is only trained on these 3 input mixtures.

In other words, we can create a brand new latent vector that are slightly outside our input training manifold: $z_{ext} = [\![ z^{n+1}, z^{n+2}, ..., z^{2*n} ]\!]$. These new input latent vectors $z_{int}$ and $z_{ext}$ can be used as input to source generator $S$. Figure 6 shows that we can prolong a 3-second audio texture to a 12-second one with a combination of interpolation and extrapolation in latent space. No direct copy and paste is used, so we naturally avoid the seam discontinuity problem.

### 3.4 CO-SEPARATION / AUDIO WATERMARK REMOVAL

Audio watermarks are commonly used in the music industry for copyright-protected audios. Supervised deep networks will require a lot of training data to learn both diverse clean audio patterns and watermark patterns for separating watermark sounds from clean sounds (Muth et al., 2018). Our DAP model can also easily generalize to handle co-separation for removing audio watermarks.

Given $K$ sounds: $S_k = S_k^* + S_{watermark}$ ($k = 1, 2, ..., K$) containing the mixture of the same unknown audio watermark $S_{watermark}$ and the clean audios: $S_1^*, S_2^*, ..., S_K^*$. The goal is to recover the clean audios. Using the proposed separation framework, we learn $K + 1$ generator networks; $K$ for the music signals and 1 for the watermark with shared network weights and input noise. As shown in Figure 7, from the $K = 3$ input mixtures, we can extract the clean music as well as the embedded watermark.

## 4 ABLATION STUDY AND DISCUSSION

**Temporal Noise Input** To capture the temporal coherence prior within audio spectrograms, we decompose individual noise inputs into temporally consistent segments. To justify the temporal noise, we compare to a variant of our model without temporal noise (w/o TN), which has no temporal consistency but with only random per-frame noise. As shown in Figure 8 (b), we can see that the model without temporal noise fails to preserve temporal coherent "basketball" sound structures while our DAP model can well capture the temporal consistent patterns in the "basketball" sound.

**Dynamic Noise** We introduce dynamic noise from curriculum learning to capture large variations in individual sounds and preserve temporal consistent structures. In particular, we use abruptly changing dynamic noise during noise input transition for better quickly learning dynamic patterns for individual sound predictions. To show the effects of dynamic noise and abrupt noise transition, we compare to two baseline models, which are without dynamic noise (w/o DN) and without abruptly changing noise (w/o AC), respectively. The results are illustrated in Figure 8 (c) and (d). We see that the w/o DN model fails to separate the two sounds and it can only capture few dynamic structures in the "violin" sound due to its weak dynamic modeling capacity. Although the w/o AC model can restore more dynamic "violin" patterns, it incorrectly adds "violin" sounds into the "basketball" sounds. In comparison, our full DAP model has strong dynamic modeling capacity for the two individual sound prediction networks immediately.

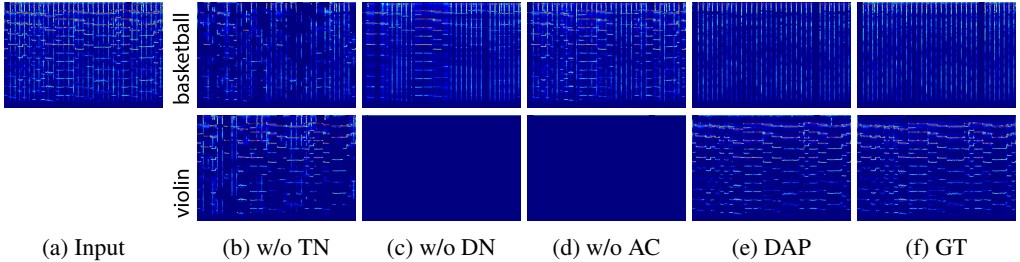

Figure 8: Ablation study on noise design. From left to right, spectrograms are from (a) input mixture (b) without temporal noise (w/o TN) (c) without dynamic noise (w/o DN) (d) without abrupt change in dynamic noise (w/o AC) (e) our full DAP model (f) ground truth.

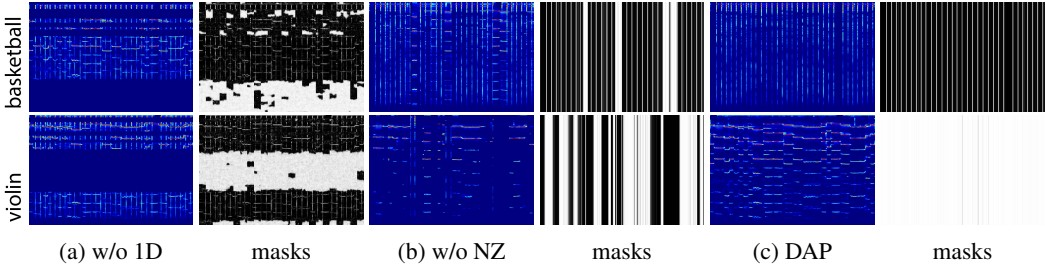

Figure 9: Ablation study on mask design. From left to right, spectrograms and masks are from models without 1D mask (w/o 1D), without nonzero mask loss (w/o NZ), and with both 1D mask and the loss (DAP).

**1D Mask Design**   Unlike commonly used unconstrained masks used in images/videos, we design a 1D mask. This design explicitly decomposes sound estimation into two sub-problems: temporal sound prediction and mask modulation, which will ease mask learning and temporal consistency modeling. Such disentanglement idea has been widely used in separating geometry and appearance estimation (Janner et al., 2017; Lin et al., 2019) To validate the effectiveness of the 1D mask, we compare it to a baseline model with unconstrained masks. As illustrated in Figure 9 (a), we can see that the model without the strong 1D mask constraint easily find a short-cut to minimize the loss functions and the unconstrained masks also capture sound content. With the 1D mask, our DAP model disentangles sound prediction and mask modulation and successfully separates the two sounds.

**Non-Zero Mask**   To demonstrate the effectiveness of the proposed nonzero mask loss, we empirically show results of our DAP without the loss during optimization in Figure 9 (b). From the mask results, we can find that mask values even in some sounding regions are close to zero for both two sounds. With the loss term, DAP can reconstruct sounds for all sounding regions.

**Discussion**   RPCA decomposes a spectrogram matrix $S$ into a low-rank matrix $S_l$ and a sparse matrix $S_s$ and $S = S_l + S_s$. To satisfy the principle of the decomposition, $S_l$ will capture repeating structures in the audio spectrogram and the remaining large variations in the spectrogram will be preserved in the sparse matrix. KAM assumes an audio source at a timestep can be estimated using its values at other nearby times through a source-specific proximity kernel, allowing addressing local redundancy in audio sources but missing modeling dynamic patterns.

In essence, various formulations have been proposed to utilize temporal redundancy and dynamics. However, it is challenging for a traditional formulation to capture the temporal consistency and dynamic patterns due to its limited capacity. To model both, we take advantage of the large capacity from deep networks and utilize temporal consistent noise inputs, which helps our network simultaneously restore temporal consistent sounds and capture large variations.

## 5 Related Work

As discussed in the introduction, our work is inspired by the recent advances in deep image prior (DIP) and double-DIP (Ulyanov et al., 2018; Gandelsman et al., 2019). Recently, Michelashvili & Wolf (2019) tried to learn a deep network prior for audio by following DIP process. Instead of learning the priors from audio signals, a predicted apriori SNR of input audio is fed into traditional denoising methods, such as LSA (Ephraim & Malah, 1985) or the Weiner filter, to perform audio denoising. Therefore their method still relied on the accuracy of existing denoising algorithms. In our deep audio prior framework, we propose to capture the inherent audio priors using deep networks without relying on or bottlenecked by the previous method.

Audio source separation is a classical problem in signal processing (Haykin & Chen, 2005; Naik et al., 2014; Makino, 2018). To address the problem, many blind audio source separation methods have been proposed, such as NMF (Virtanen, 2007; Smaragdis & Brown, 2003), RPCA (Huang et al., 2012), and KAM (Liutkus et al., 2014; Yela et al., 2018). For multichannel audios, independent component analysis (ICA) is also commonly used (Hyvärinen & Oja, 2000; Smaragdis, 1998; Dinh et al., 2014); in our paper, we focus on separation from single-channel audios. To improve separation performance, supervised NMFs are explored in (Mysore & Smaragdis, 2011; Smaragdis, 2006). However, these methods usually have limited capacity to handle various and complicate sound patterns. In our work, we leverage the large capacity from deep neural networks to encode both the temporal audio priors and large variances that exist in real-world complex audios.

Recently, deep audio separation networks are proposed (Chandna et al., 2017; Hershey et al., 2016; Isik et al., 2016; Chen et al., 2017; Smaragdis & Venkataramani, 2017; Le Roux et al., 2015; Wang & Chen, 2018), all of which require a large amount of audio training data. Also, a deep model trained on a certain type of sound sources cannot generalize well on audios from unseen categories, as we shown in §3.1. These limitations restrict the existing deep networks to address scalable universal audio separation. Unlike the existing approaches, our DAP framework is only trained on the single input mixture audio, does not require any additional training data, and hence is immune from the data mismatch between training and testing sets.

## 6 Conclusion and Future Work

We have introduced Deep Audio Prior, a new audio prior framework that requires zero training data. Thanks to the universal and unsupervised nature, we are excited about the potential applications that DAP can enable. We have demonstrated impressive results on challenging tasks, even when comparing our model to models trained on a large amount of supervised data. All of our examples are listed on the anonymous webpage: `https://iclr-dap.github.io/Deep-Audio-Prior/`

**Limitations and Future Work**  Our framework naturally extends to multiple sound sources, e.g., co-segmentation with 4 sources in total. Yet for audio recorded in the wild, one main challenge is to decide how many effective sound sources are present (Girin et al., 2018). One direction that we would like to explore is to use the output/error metrics from DAP separation to iteratively decide the optimal number of effective sources. Visual information can also be used to guide this process (Sodoyer et al., 2002; Aytar et al., 2016). It remains an open challenge on how to design a *deep audiovisual prior* to combine audio and visual information to distill a more robust representation.

In the interactive editing demo (§3.2), we show that the progressive refinement based on user input can be easily done within seconds. However, the initial DAP separation usually takes in the order of minutes for a short audio segment. Possible acceleration in the initial training stage is a promising direction. In supervised deep learning, distilling knowledge from multiple models has shown great success (Hinton et al., 2015). Given that DAP works for audios in the wild, we are interested in how to robustly distill useful information from a large amount of learned generators from many single audios.

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

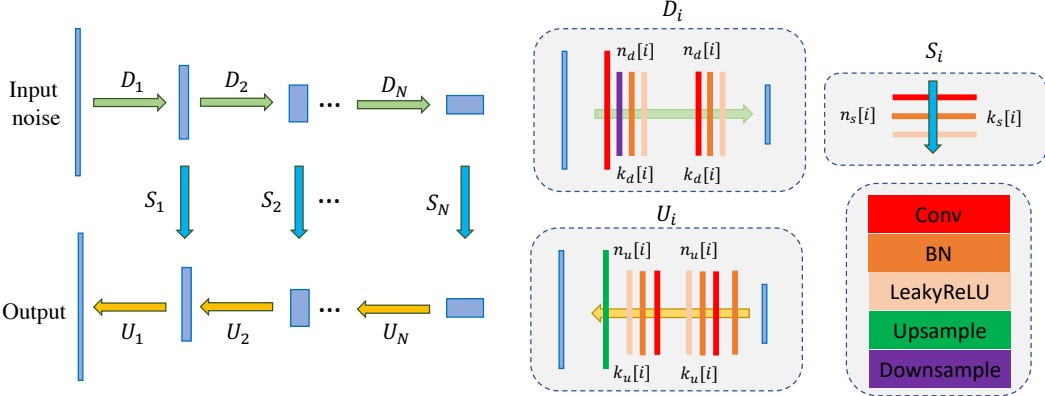

Figure 10: Our audio generator and mask generator networks share the same U-Net structure as in (Ulyanov et al., 2018). There are mainly three modules: downsampling module (see $D_i$), upsampling module (see $U_i$), and skip connection module (see $S_i$). Batch normalization (BN) (Ioffe & Szegedy, 2015) is used accelerating and stabilizing training and LeakyReLU (He et al., 2015) is used as the nonlinearity function. $n_d[i]$, $n_u[i]$, $n_s[i]$ denote the number of filters at depth $i$ for the downsampling, upsampling, and skip-connections respectively. The values $k_d[i]$, $k_u[i]$, $k_s[i]$ correspond to the respective kernel sizes for the convolutional layers in different modules.

Table 2: Better sample ratio (%) comparing with NMF/RPCA/KAM in terms of numerical metrics: SDR/SIR/LSD on Universal-150. All results are larger than 50%, which shows that our DAP can achieve better results than the compared methods on a majority of testing samples.

|     | DAP vs. NMF | DAP vs. RPCA | DAP vs. KAM |
|-----|-------------|--------------|-------------|
| SDR | 75.3        | 66.0         | 56.0        |
| SIR | 76.7        | 62.7         | 66.7        |
| LSD | 71.4        | 58.0         | 71.4        |

## A  APPENDIX

### A.1  NETWORK STRUCTURE

We use an UNet architecture as in Ulyanov et al. (2018) for our audio and mask generator networks. In the networks, the downsampling is achieved by setting the stride of the convolutional layer as 2, and the upscaling is implemented by Bilinear interpolation. In our implementation, we use a relative smaller UNet structure, which only consists of $N = 3$ downsampling modules and upsampling modules. The parameters of the used UNet are $n_d = n_u = [16, 32, 64]$, $k_d = k_u = [5, 5, 5]$, $ns = [0, 0, 4]$, $ks =$[None, None, 1]. During tuning parameters of the UNet, we have the same observation as Ulyanov et al. (2018) that a wide of range of hyper-parameters for the UNet can achieve similar performance. We empirically found that our models can be converged in 5000 iterations.

### A.2  ADDITIONAL SOUND SEPARATION RESULTS

To further clarify the numerical results in the Table 1, except from the mean SDR, mean SIR, and mean LSD, which are sensitive to over large or over small values (variance also has the same issue), we count the better sample ratio based on the three metrics. The ratio will count our DAP is better than the compared the method on how many testing samples. For example, compared to the NMF in terms of SDR, the ratio is 0.753, which shows that our method is better than the NMF on 75.3% testing samples. The comparison results are shown in Table 2. We can see that our DAP achieve better results on a majority of testing samples comparing to NMF/RPCA/KAM.

To further validate our DAP on sound source separation, we compare with other methods on a standard sound separation benchmark from (Vincent et al., 2007), which has 20 testing samples and each sample has 3 clean sounds. To evaluate different sound separation methods, we take the first two sounds from each sample to compose 2-sound mixtures. As shown in Table 3, We can see that

Table 3: Comparison between DAP/NMF/RPCA/KAM in terms of numerical metrics: SDR/SIR/LSD on a standard source separation benchmark (Vincent et al., 2007). For SDR/SIR, higher is better and for LSD, smaller is better. The best results on the three metrics are from our method, DAP.

|  | NMF (Spiertz & Gnann, 2009) | RPCA (Huang et al., 2012) | KAM Yela et al. (2018) | Proposed DAP |
|---|---|---|---|---|
| SDR | -3.36 | -4.49 | -2.16 | **-1.37** |
| SIR | -0.93 | -1.00 | -0.32 | **0.82** |
| LSD | 0.56 | 0.57 | 2.19 | **0.48** |

Table 4: Better sample ratio (%) comparing with NMF/RPCA/KAM in terms of numerical metrics: SDR/SIR/LSD on the source separation benchmark (Vincent et al., 2007). Most ratio results are larger than 50%, which shows that our DAP can achieve overall better results than the compared methods.

|  | DAP vs. NMF | DAP vs. RPCA | DAP vs. KAM |
|---|---|---|---|
| SDR | 60.0 | 70.0 | 50.0 |
| SIR | 55.0 | 40.0 | 55.0 |
| LSD | 70.0 | 65.0 | 95.0 |

our method outperforms the three compared methods in terms of SDR, SIR, and LSD. The Table 4 illustrates better ratio results. We can also find that our DAP still achieves overall better results.

## A.3 DYNAMIC RESULTS

To model dynamic patterns with large variations in audio sources, we we gradually add dynamic noise into inputs as we progress more training iterations. In our implementation, the two iteration parameters: $T_1$ and $T_2$ in the Equation (6) are set as 2000 and 4000, respectively. Note that the maximum training iteration number is 5000. When iteration number is smaller than $T_1$, only temporal coherent noise inputs are used; when iteration number is in [2000, 4000], we gradually add dynamic noise into inputs; when iteration number is large than 4000, noise inputs are fixed. To illustrate the training dynamics, we show separation results at different training iterations in Figure 11. We can see that our model first restore temporal coherent audio sources with the input noise constraint; when we introduce dynamic noise into input (after 2000 iterations), the model will quickly learn the dynamic patterns in the sound mixtures and well capture sound variations while preserving temporal consistent audio sources.

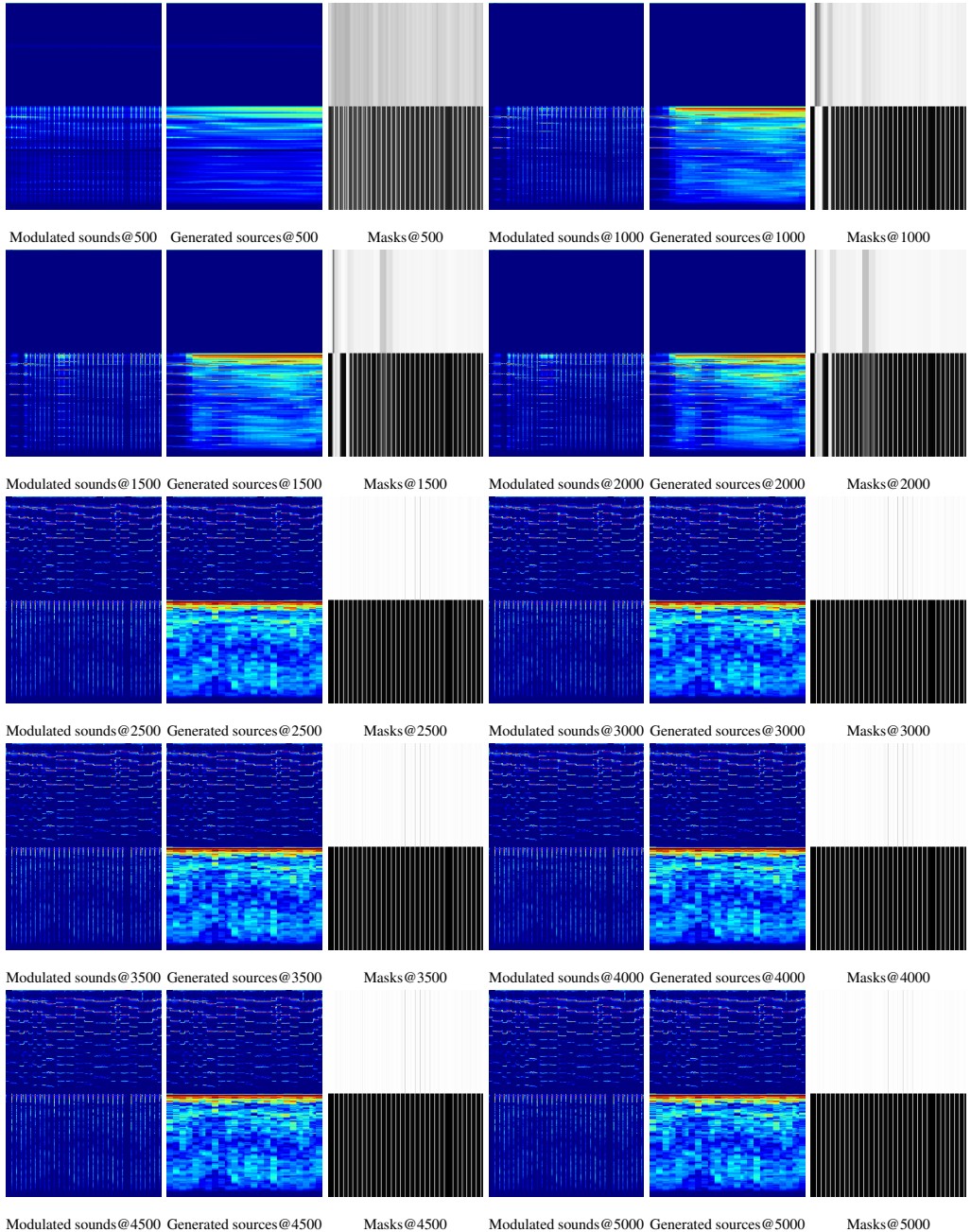

Figure 11: Dynamic results of our DAP from different training iterations. We show generated audio sources, masks, and the separated sounds with corresponding mask modulations at 500*th*, 1000*th*, ..., 5000*th* iterations, respectively. After adding dynamic noise into inputs at 2000*th* iteration, the model will quickly capture the violin sounds with large variations (see results at 2500*th*).

