# OpenReview forum: "Deep Audio Prior"
_ICLR.cc/2020/Conference — Reject_

### Official Review · AnonReviewer2 · 2019-10-21
**Official Blind Review #2**

**Rating:** 3

**Review:**

This paper presents a method for blind source separation relying on randomly initialized networks to decompose an input audio spectrogram into two components.
The networks are designed to promote temporal contiguity of spectral energy in the estimated signals, which are modulated (in time) by estimated masks.
The proposed method is evaluated on a collection of 150 random mixes of sounds, and performs favorably relative to some standard baseline methods (RPCA, NMF, KAM).

This seems like a promising line of work, but at this point I think the weaknesses of the paper outweigh its strengths (as detailed below).  Some of these points may be addressed during discussion, but I currently lean toward reject.


Strengths of the paper:

- The ideas are interesting, and appear to perform well on a simulated and real(ish) data.

- The authors investigate several variations and applications of source separation, including interactive editing, co-separation, and texture synthesis.

- A small (qualitative) ablation study is included to clarify the importance of different components of the loss function.


Weaknesses of the paper:

- Much of the presentation is vague or opaque.  There is little detail provided about the specific architectural parameters of the model, and the diagram (figure 1) does not appear to match the equations.  Specifically, it's unclear whether S_1^* is a function of S_mixture or not.

- The quantitative evaluation focuses entirely on one (new) dataset with unclear characteristics.  No details are provided about the evaluation protocol, and in particular, the tuning of hyperparameters for the various methods under comparison.  Aggregate statistics are included (mean? SDR, etc), but no notion of variance or error bars are included.  It's not ultimately clear how fair this evaluation is.  There should at minimum be a comparison on a standard source separation or speech enhancement dataset, in addition to the new set presented here.

- Most of the spectrogram figures appear to be upside-down, which is confusing.  The details of the audio processing are omitted: STFT parameters are stated, but not the sampling rate.

- There are numerous typos ("grounth", "spectrogram stokes", etc), indicative of the authors not running a spell-checker on their submission.


Questions for the authors:

- The model itself consists of several competing loss functions, but it seems like they may have a trivial, optimal solution at S1 = S_mix (M_1 = 1) and S_2 = 0 (M_2 = 0 or 1).  As far as I can tell, this solution would trivially minimize each term of equation 9.  (If S_1* does not depend on S_mixture, this may be less of an issue, but the trivial solution may still exist when the driving noise is of sufficiently high dimension and the optimization is run long enough.)  Am I misunderstanding the algorithm, or is there some deeper reason why this solution would not be preferred?


**Experience Assessment:**

I have published one or two papers in this area.

**Review Assessment: Checking Correctness Of Derivations And Theory:**

I assessed the sensibility of the derivations and theory.

**Review Assessment: Checking Correctness Of Experiments:**

I carefully checked the experiments.

**Review Assessment: Thoroughness In Paper Reading:**

I read the paper at least twice and used my best judgement in assessing the paper.

---

> ### Author Response · Authors · 2019-11-10
> **Responses to Official Review #2 (Part 1)**
>
> Thank the reviewer for recognizing our work and providing constructive comments. We try to address the concerns brought up below.
>
> A1: Since U-net is commonly used in previous works (Ulyanov et al., 2018; Gandelsman et al., 2019), we only give the reference in the paper for making the paper be concise and leaving pages for delivering methods and several exciting applications. But now, to make it more clear to readers, we have illustrated the network in the updated appendix (please see Section A.1 and Figure 10).
>
> Figure 1 is consistent with Equation 1 in the paper. As shown in the Figure and formulated in the Equation, we decompose a sound mixture to individual sounds with corresponding mask modulations. To achieve the decomposition, unlike all existing sound separation networks that use sound mixtures as separation network inputs, our DAP model learn individual audio source generator network and mask generator network for each audio source with randomly sampled noise as inputs. Therefore, S_{mixture} is not a function input and it will serve as the groundtruth signal for learning as in Equation 2.
>
> A2: We have indeed faced the problem in our evaluation to find an existing dataset for validating our method whether can learn to handle various audio sources. However, we learn that most existing datasets only contain either music or speech sounds. To evaluate the generalizability of our deep audio prior network, we build a dataset containing 150 sound mixtures covering a large range of sound categories appeared in our daily life.
>
> For evaluation, we adopt the commonly used mir_eval library https://github.com/craffel/mir_eval to compute SDR and SIR scores and LSD is implemented by us. For the compared methods, we carefully tuned their parameters for fair comparisons. However, the conventional matrix decomposition and kernel regression-based blind separation methods have limited capacity to well model rich and diverse audio patterns as shown in Figure 3 and also results in our anonymous Github repository.
>
> Since averaged scores of SDR, SIR, and LSD are commonly used metrics in previous literature, we also adopt the practice in our paper. For additional comparison, we have compared our method with recent speech enhancement methods: DNP and SEGAN on a noisy speech, where the clean speech is from the used dataset in the SEGAN paper.
>
> To further address the concerns, we would like to show additional results on our dataset using a new metric and test our method on an existing standard sound separation dataset containing music and speech sounds. We have included the additional experimental results in the Appendix of the paper (please see Section A.2.)
>
> To further clarify the numerical results, except from the mean SDR, mean SIR, and mean LSD, which are sensitive to over large or over small values (variance also has the same issue), we count the better result ratio based on the three metrics. The ratio will count our DAP is better than the compared the method on how many testing samples. For example, compared to the NMF in terms of SDR, the ratio is 0.753, which shows that our method is better than the NMF on 75.3% testing samples. In terms of SDR, SIR and LSD, compared to NMF, RPCA, and KAM, the ratio results are (0.753, 0.66, 0.56)@SDR, (0.767, 0.627, 0.667)@SIR, and (0.714, 0.58, 0.714)@LSD, respectively. The results further show the superiority and robustness of our method.
>
> To further validate our method, we compare with other methods on a sound separation benchmark (Vincent et al, 2007), which has 20 testing samples and each sample has 3 clean sounds. In our testing, we take the first two sounds from each sample to compose sound mixtures. The (SDR, SIR, LSD) results for NMF, RPCA, KAM, and DAP are (-3.36, -0.93, 0.56), (-4.49, -1.00, 0.57), (-2.16, -0.32, 2.19), and (-1.37, 0.82, 0.48). We can see that our method outperforms the three compared methods in terms of SDR, SIR, and LSD. Note that it is smaller and better for LSD.
>
> E. Vincent, R. Gribonval and M.D. Plumbley, Oracle estimators for the benchmarking of source separation algorithms, Signal Processing 87(8), p. 1933-1950, 2007.

---

> > ### Author Response · Authors · 2019-11-10
> > **Responses to Official Review #2 (Part 2)**
> >
> > A3: For spectrogram visualization, we adopt a publicly available code from a recent ECCV paper (Zhao et al, 2018). In our experiments, the audio sampling rate is set as 11000 and we have added it into the updated paper.
> >
> > Zhao, Hang, et al. "The sound of pixels." Proceedings of the European Conference on Computer Vision (ECCV). 2018.
> >
> > A4: Thanks for the correction. We indeed carefully checked our paper and even used a professional software, Grammarly, to go through the full paper. We are sorry we missed 4 typos ("illustrared, initialily, grounth, stokes") in the submission. We have fixed them in the revised paper.
> >
> > A5: For most distinct audio sources, they will contain highly different time-frequency patterns. Since we have temporal consistent noise as inputs for the early training iterations (please check equations (3), (5), and (6)), which will enforce two audio generators to generate temporal consistent audio patterns for individual sources rather than mixed sources with larger temporal variations.
> >
> > However, a model with only temporal consistent noise can not well reconstruct time-frequency variations. We gradually add dynamic noise into the input to address the issue, however, the dynamic noise not only helps model variations but might also facilitate the trivial solution for certain cases. For example, as shown in Figure 8, the basketball sound contains highly temporal coherent patterns, but the violin sound has massive dynamic patterns (only one highly temporal coherent source in the mixture). With only temporal consistent noise, one audio generator can well restore the temporal coherent basketball sound but the dynamic violin patterns can not be well modeled by the two audio generators. To capture variations in sounds, we first gradually add smoothly changing dynamic noise. However, we found that the diligent audio generator who restores the basketball patterns gradually adds variations to the source. The other audio generator keeps be lazy and only generates zeros (see Figure 8(d)). To make the lazy audio generator be active, we empirically found that dynamic noise with an abrupt transition is pretty efficient, which can immediately equip variation modeling capacity into two audio generators. Thus, the two audio generator networks will quickly add variations into individual sources while preserving temporal consistent patterns (see Figure 8(e)).

---

### Official Review · AnonReviewer3 · 2019-10-24
**Official Blind Review #3**

**Rating:** 6

**Review:**

The authors detail a set of priors for unsupervised decomposition of individual spectrograms into their component parts. The introduce reasonable constraints on temporal coherence (consistency and dynamic shifts) and mask activations (at least one component always activated). They also regularize sources to not overlap spectrotemporally. Decomposition is performed by training the weights of a U-Net on a single spectrogram as in deep image priors. The authors demonstrate quantitative improvements on blind source separation over other data-agnostic techniques, and qualitative use of the model for interactive editing, audio texture synthesis, and audio watermark removal. The work also performs an ablation study to qualitatively demonstrate the importance of each element for the prior. The experiments are performed well and explained clearly. They also introduce a dataset of diverse mixtures for future comparisons.

Pros:
* Important motivation for why audio has different properties than images (even if it can be represented as a "image" spectrogram). The priors are well-motivated by the dynamics of audio.
* Good ablations and quantitative comparisons to baselines.

Cons:
* Some details could be better demonstrated / explained (even if only in the appendix). For example the paper cites the network architecture, but a local description would be helpful. Similarly, the latent dynamics are carefully regularized, so visualizing them would be helpful to understand the dynamics.
* The scaling of the technique is not supported by the current experiments. The authors claim they have extended to 4 sources, but all experiments in the paper seem to only involve two sources.
* More motivation could help in terms of the value of non-amortized methods like deep priors, vs. other approaches such as pretraining or self-supervised methods. While it is difficult to get lots of labeled data for a specific task, the argument was not convincingly made that methods like deep priors should outperform methods that use pretrained priors on adjacent tasks (where collecting data is easy).

**Experience Assessment:**

I have published in this field for several years.

**Review Assessment: Checking Correctness Of Derivations And Theory:**

N/A

**Review Assessment: Checking Correctness Of Experiments:**

I carefully checked the experiments.

**Review Assessment: Thoroughness In Paper Reading:**

I read the paper at least twice and used my best judgement in assessing the paper.

---

> ### Author Response · Authors · 2019-11-10
> **Responses to Official Review #3**
>
>  We would like to thank the reviewer for the constructive feedback. We address the concerns from the reviewer as follows:
>
> A1: Thanks for the suggestion! We have updated our paper with an appendix to show the used U-net structure (please see Section A.1 and Figure 10) and visualize the results from different training iterations with the proposed dynamic noise inputs to help readers better understand the dynamics (please see Section A.3 and Figure 11).
>
> A2: By "co-segmentation with 4 sources in total", we refer to the audio watermark removal application. In the application, we successfully separate four different audio sources with co-separation using our deep audio prior network. We will clarify further this in the revision.
>
> A3: This is a really good question! We indeed have observed a lot of works on self-supervised audio learning taking the natural audio-visual synchronization as supervision. The pre-trained models have been proved to be useful in many downstream tasks, such as audio classification, audio event detection, and sounding scene localization. But when the existing methods want to extract clean audio sources from sound mixtures using the pre-trained models, they usually need to access clean single-source sounds for composing training sound mixtures and then generating ground truth masks for supervised separation learning. Unlike the existing methods, with the proposed deep audio prior, our network can directly learn from a single sound mixture without requiring clean single-source sounds.
>
> Another motivation is to use the individually learned models to distill a more robust combined model. Since each individual model is cleaner and carries semantic representations (i.e. temporally coherent), in addition to the 4 applications that we showed in the paper, we believe this could have more downstream impact for audio synthesis, audio representation, etc.

---

### Official Review · AnonReviewer1 · 2019-10-26
**Official Blind Review #1**

**Rating:** 6

**Review:**

This paper introduces deep audio prior (DAP), which uses CNN'S prior to perform classical tasks in audio processing: source separation, denoising, texture synthesis, co-separation. This paper gets inspiration from deep image prior and adapts it to audio, by introducing a lot of insights in the audio domain, which I believe is a good amount of contribution.
I have the following questions on the paper:
(1) The source generation part is a bit confusing. When testing on real-world examples, do you need to generate sources? I want more explanations on the ablations studies in Section 4, temporal/dynamic sources. What are the input/output, how is the noise generated, how is the model trained.
(2) In general, I expect more details in the paper, like the model architecture (does that affect performance), how training is performed (iteration, convergence, etc).
(3) Some notations are missing, e.g. equation (4).

**Experience Assessment:**

I have published in this field for several years.

**Review Assessment: Checking Correctness Of Derivations And Theory:**

I assessed the sensibility of the derivations and theory.

**Review Assessment: Checking Correctness Of Experiments:**

I carefully checked the experiments.

**Review Assessment: Thoroughness In Paper Reading:**

I read the paper thoroughly.

---

> ### Author Response · Authors · 2019-11-10
> **Responses to Official Review #1**
>
> We would like to thank the reviewer for carefully reading our paper and appreciating the contributions made by our work.
>
> A1: When testing on real-world examples, our method generates both a sound spectrogram and its corresponding mask modulator. The input latent variables are randomly sampled noises. The workflow for the real-world audio separation is the same as synthetic audio separation.
>
> Temporal source: For i-th source in a sound mixture, we use randomly sampled temporal coherent noise z_i (please see Equation (3) in the paper) as input for the audio generator network S_i to generate a temporal consistent audio source S_i(z_i) and we use randomly sampled Gaussian noise g_i as input for the mask generator M_i to generate a mask M_i(g_i) to modulate the generated corresponding audio source S_i(z_i) to obtain the separated i-th source S_i(z_i)*M_i(g_i). For the temporal noise only model, once we randomly sample the input noises, they will be fixed during training.
>
> Dynamic source: To preserve the temporally consistent patterns and also hallucinate dynamic patterns in audio sources, we use dynamic noise z_i (please see Equation (5) in the paper) as an input during training for the audio generator network S_i. This is the only difference between the dynamic source generation model and the temporal consistent noise only model. For the dynamic source generation model, the input noises for the audio generator networks will be changed for different training iterations as defined in Equations (5) and (6).
>
> A2: Thanks for the suggestion. Since we used an U-net architecture as in deep image prior (DIP) (Ulyanov et al., 2018), we only give a reference for making the paper be concise and leaving pages for describing methods and several applications.  Now, to make it more clear to readers, we have illustrated the used U-net in the updated appendix (please see Section A.1 and Figure 10).
>
> For the model architecture, we also have the same observation as in (Ulyanov et al., 2018) that the wide range of hyperparameters and architectures give similarly acceptable results. For training, we empirically found that the models can achieve a good convergence with 5000 iterations in our experiments. We will include the variation structure we tested in the revision.
>
> A3: Thanks for pointing it out. The S in Equation (4) refers to the generated audio spectrogram for the individual audio sources. We have added the notations to make it more clear.

---

### Author Response · Authors · 2019-11-10
**Paper Revisions**

We appreciate all the reviewers recognizing the novelty of our work and providing helpful suggestions! We have made some modifications to address the concerns raised by the reviewers. The main changes in the paper include:

(1) We illustrate the used U-Net architecture in Figure 10 and describe the details of the network in the Appendix Section A.1. We use a UNet architecture as in (Ulyanov et al., 2018) for our audio and mask generator networks. In the networks, the downsampling is achieved by setting the stride of the convolutional layer as 2, and the upscaling is implemented by Bilinear interpolation. In our implementation, we use a relative smaller UNet structure, which only consists of 3 downsampling modules and upsampling modules. For more network details, please kindly see the Appendix Section A.1 and Figure 10.

(2) To address the concerns from reviewer #2. Except from the aggregated statistic, we use a new metric to measure a better sample ratio, which is not sensitive to over large or over small results on certain testing samples. In addition, we compare our DAP with other methods on an additional standard sound separation dataset. The numerical results are shown in Tables 2, 3, and 4 of the Appendix and we discuss the experiments in Appendix Section A.2. The additional results further validate the effectiveness of the proposed DAP.

(3)We show dynamic results of our DAP at different training iterations in Figure 11 and discuss it in the Appendix Section A.3.

(4) 4 typos ("illustrared, initialily, grounth, stokes") are fixed and the audio sampling rate 11000Hz is provided, and Equation 4 adds more clear notations.

---

### Decision · Program_Chairs · 2019-12-19

**Decision:**

Reject

**Comment:**

This paper proposes to use CNN'S prior to deal with the tasks in audio processing. The motivation is weak and the presentation is not clear.  The technical contribution is trivial.